# The Physiological Experimental Study on the Effect of Different Color of Safety Signs on a Virtual Subway Fire Escape—An Exploratory Case Study of Zijing Mountain Subway Station

**DOI:** 10.3390/ijerph17165903

**Published:** 2020-08-14

**Authors:** Na Chen, Ming Zhao, Kun Gao, Jun Zhao

**Affiliations:** School of Mechanics and Safety Engineering, Zhengzhou University, Zhengzhou 450001, China; zm917204851@gs.zzu.edu.cn (M.Z.); kungup@163.com (K.G.); zhaoj@zzu.edu.cn (J.Z.)

**Keywords:** safety sign color, subway fire escape, virtual reality, physiological experiment, eye-tracking

## Abstract

Safety signs play a very important role in people’s evacuation during emergencies. In order to explore the appropriate color for subway safety signs, four safety signs of different color combinations are designed, and the virtual reality, eye-tracking technology, and physiological indicator measurement are used in a virtual subway fire escape experiment. A total of 96 participants with equal distribution in gender and four different color combination groups were recruited. Participants’ eye-tracking and physiological data (heart rate, skin conductance) were real-time recorded through ErgoLAB V3.0 in the whole experiment. The relationship between Color_of_safety_sign and escape performance, eye-tracking indicators, and physiological indicators is discussed respectively through SPSS. The results show that “Green and black” group has the best evacuation escape performance, low cognitive load, high search efficiency on safety signs, and the highest stress level and immersion and “Green and black” can be the most appropriate color for safety sign. This research is of certain significance for improving the function of subway fire-fighting infrastructure and the resilience of the metro system. Moreover, it can provide references and advice on risk management, emergency evacuation, and so on.

## 1. Introduction

The subway has become one of the main types of urban transportation, due to its capability to accommodate large volume of passengers, fast speed, and high efficiency. Due to many factors, there are lots of risks on the construction and operation of subways, such as construction risk [1,2], crowded stampede risk [3], fire risk, etc. In particular, subway fire can cause a large number of casualties and property damage. Zhang et al. [4] mainly classified the hazards of fire into three categories: heat, toxic gases, and smoke in his numerical simulation. Inhalation of high temperature gas may cause edema of tracheal mucosa and lead to death; excessive inhalation of toxic gases can cause coma and even death [5]; smoke from the fire worsens the situation and causes choke and poor visibility [6]. Illuminated emergency safety signs are effective tools to help people evacuate to safe areas in the event of an emergency in a building. However, safety signs are green in China while safety signs may be red or green in USA, so what color safety signs should be?

According to the definition of “resilience”, “resilience” means the ability of a system to adapt, reduce, and recover from various risks [7]. Therefore, the study of safety sign color is conducive to making up for the defects of fire protection systems [8], improving the functional construction of fire protection infrastructure, and thus improving the fire-fighting capability of the system [9].

There have been a lot of studies on safety signs in the world. In terms of the presence or absence of safety signs, the presence of safety signs is more conducive to escape [10]. In general, the visibility of a sign increases with the size of the sign [11]. In terms of the color of the safety sign, Wong’s [12] study showed that green exit signs had higher visual performance than other color. One interesting study showed that while participants regarded the color red as a safety sign in a questionnaire, they walked out of a green door in a virtual experiment; the results challenged the local exposure hypothesis and supported the semantic association hypothesis [13]. From the dynamic and static points of view on the safety sign, a dynamic signage system combined with dynamic features has more visibility than a static signage system [14,15]. Flashing lights and strobe lights increased the use of an emergency exit compared to the standard emergency exit design [16,17], especially safety signs combined with green flashing lights may guide people to safety better than safety signs combined with red or orange flashing lights. Similarly, another research showed that green or white flashing lights performed better than blue lights, and a flashing rate of 1 and 4 Hz performed better than a flashing rate of 0.25 Hz [18]. The ADSS (Active Dynamic Signs System) proposed by Galea et al. [19,20] and VMS (Variable Message Signs) System proposed by Ronchi et al. [21] have been proved to be effective in improving the escape efficiency through experiments. Olander et al. [14] proved through experiments that placing a red “x” mark on the entire exit sign can convey a clear message of dissuasion. However, Kwee-Meier et al. [22] thought it was dangerous to use combined prohibition–recommendation signs in the real evacuation process.

Most studies have shown that green would be the most suitable safety sign color. Daniel’s study showed that green light should be used at emergency exits [16]. Kinateder et al. found that most students walked toward green signs [13]. In Wong’s research, green exit signs had the highest visual performance [12]. However, in most of the above researches, few studies have analyzed the safety signs from eye-tracking data or physiological indicator changes. In addition, in most of the existing studies, face-to-face questions, questionnaires, or video analysis [23,24] were used to judge whether the participants saw the safety signs, which result in much error. New technologies, such as VR (virtual reality) and ET (eye-tracking) technology, can reduce error and improve reliability.

VR is widely used in many fields due to its “3I” (Immersion, Imagination, Interaction) features, which can reduce cost consumption [25] and make it safer [26]. Under the premise of not being reminded to pay attention to dynamic and static signs in virtual reality, static and dynamic signs had the same effect on changing people’s behavior [27]. Therefore, this study is based on virtual reality technology under the condition of static signs without reminding participants to pay attention to safety signs.

ET technology is a useful tool for assessing the effectiveness on signage. Occhialini et al. [28] believes that ET can be used as a tool for exploring the behavior. Vilar et al. [29] proposed that eye-tracking technology may be used to verify how safety signs influence on navigational behavior. Meißner et al. [30] believed that virtual reality technology and eye-tracking technology can bring new research opportunities for identifying the usability of virtual reality environment. Tang et al. [31] used ET and VR technology to study human behavior in urban subways during peak hours. This study will also use virtual reality and eye-tracking technology to explore the effect of different color safety signs on subway fire escape performance. EDA (Electrodermal Activity) [32,33,34], and HR (Heart Rate) [32,34] have become indicators reflecting individual stress levels. These indicators will be used to explore the impact of the different color of safety signs on personal stress levels under a virtual subway fire escape scene in this study.

## 2. Materials and Methods

### 2.1. Case Study

There are few studies on the color of safety signs, especially in subways. As a preliminary exploratory experimental study, a single case study [35] method was used in this research, which could provide basis for the future multiple case studies [36]. Currently, the color combination of safety signs in Chinese related national standard is ”Green and white”, the real visual effect after charging is “Green and black” (see Figure 1), and all the color combination, design, and the real visual effect of the safety signs are all the same in the subway stations in each city in China. Therefore, we only chose one subway station, Zijing Mountain station in Zhengzhou, China, as a case study.

Zijing Mountain station (see Figure 2) is the largest transfer subway station of Line 1 and Line 2 in Zhengzhou, China, with a high passenger flow each day. There are four floors in it: −1F (minus 1th floor) is the station hall, −2F is the transfer floor shared by Line 1 and Line 2, −3F is the platform of Line 1, and −4F is the platform floor of Line 2. Here we take Zijing Mountain subway station as a case study, using SketchUp and Unity3D to construct the virtual subway fire escape scene, according to the real Zijingshan subway station in Zhengzhou, China, with 1:1 scale (see Figure 3).

### 2.2. Virtual Scene Settings

(1)The design of four different color combinations of safety signs

According to the standards in Table 1, we suggested and designed four different color combinations of safety signs, that is, “Green and black”, “Red and white”, “Yellow and black”, and “Blue and white”, as shown in Figure 4. The safety sign color combination in Chinese GB 13495.1-2015 and GB/T 23809-2009 is “Green and white” (see Figure 1a), but the real visual effect after charging is like “Green and black” in Zhengzhou Zijing Mountain subway station (see Figure 1b) and other subway stations in China. Therefore, we chose “Green and black” as control group, and the other three color combinations as experimental groups. In the four groups, except for the different color of safety signs, all the other experimental scene’s construction, design, settings, and experimental procedures are exactly the same. Moreover, each group has the same number of participants and gender ratio (male:female = 1:1), not a single sex composition [37].

(2)The illumination and brightness Settings

Studies have shown that people tend to go to a wider and brighter place [44]; therefore, in order to avoid the influence of environmental brightness on the experimental results, the illumination is evenly set in the virtual scene. At the same time, considering the influence of smoke in the fire, the brightness of the background light is set to 50% of the normal brightness.

(3)Moving speed Setting

In a virtual tunnel escape experiment, Cosma et al. [45] set the moving speed of the virtual character to be 1.2 m/s, and Kinateder et al. [46] set a fixed speed of 2.86 m/s for participants to move with a small joystick on a gamepad in their virtual tunnel fire evacuation experiment. Lin et al. [33] set a constant speed of 1.5 m/s for manipulating the joystick in a virtual indoor way-finding performance experiment. Through several trials, here we set a constant speed of 1.5 m/s to move or make turns in the virtual subway fire escape with a HTC trackpad in hand, so as to avoid moving too slow or too fast (which may cause motion sickness).

(4)Other Settings

Considering the VR motion sickness, the VR helmet was worn for no more than 20 min during the experiment [27]. If the participants did not succeed in escaping in 20 min, the failure of escape would be recorded. In this study, no other NPCs (Non-Player Characters) were added, so as to avoid the interaction between people [46,47,48], and the evacuation movement in the group situation was also not discussed [24,49]. The whole evacuation process was completely dependent on the individual ability and did not include the impact of other humans or group factors on it, which was an active evacuation mode [50].

In the formal experiment, participants were told to escape to the ground exits as soon as possible when a fire began in the subway. Participants were never prompted to pay attention to any safety signs throughout the whole experiment. The escape starting point was the symmetrical position of the most unfavorable floor [51], that is, the minus 4th floor (−4F), and the end points were the ground exits. The experimental site was located in the safety ergonomics Lab in the school of mechanics and safety engineering, Zhengzhou University, in China. There was no environmental noise in it, and everything was suitably arranged to meet the experimental requirements.

### 2.3. Participants

A total of 96 undergraduate or graduate students (mean age = 22.15 ± 2.14 years, ranging from 18 to 28 years old), with equal distribution in gender and four different color groups, that is, twenty-four participants (12 females and 12 males) in “Green and black”, “Red and white”, “Yellow and black”, and “Blue and white”, respectively, from Zhengzhou University in China, were recruited for the study, and all of them signed the experiment informed consents. All participants had normal or corrected-to-normal vision, as well as normal color vision. Besides, all participants had no hypertension, diabetes, nor any known heart or respiratory disease [52] and also had no strenuous exercise or caffeine intake on the day of and before the experiment. All participants got paid for their participation.

### 2.4. Apparatus

The main hardware and software used in the experiment are shown in Table 2.

### 2.5. Experiment Design and Procedure

Before the experiment, participants needed to fill out an informed consent form, then completed a pre-experiment questionnaire, which mainly included participants’ basic information (gender, age, major, etc.), familiarity with Zijing Mountain station, and an adapted Positive and Negative Affect Schedule (PANAS) [53]. It was expected that it would take 3 min/person to fill in the pre-experiment questionnaire.

Practice phase. In this phase participants would stroll in a virtual sightseeing scene, which is not related to the experiment (see Figure 5). Through this, participants would exercise the basic operation of the HTC trackpad, to move or turn round. It was expected this phase would take 2–4 min/person.

Formal experimental phase. First, after participants rested to be stable, the physiological detectors of EDA and PPG would be put on them. The EDA sensors were placed on each participant’s index and middle fingers, in good contact with the skin, and the PPG sensor was clipped to each participant’s earlobe (see Figure 6a). Then, participants’ baseline physiological data were real-time recorded for 2 min. Next, participants would wear HTC Vive Pro Eye (see Figure 6b) to start the escape in the virtual subway fire (see Figure 6e–g) with the HTC trackpad in hands (see Figure 6b). Before and during the entire virtual fire escape, participants were not reminded to pay attention to any safety signs. Meanwhile, participants’ physiological data and eye-tracking data were real-time recorded, by ErgoLAB V3.0 man-machine-environment synchronous cloud platform, in the entire baseline period and the whole fire escape period (see Figure 6c–f). After the escape experiment, participants were asked to fill out a post-questionnaire, which mainly included an adapted PANAS [53], the understanding of the meaning of the four different color safety signs, and suggestions for the entire experiment.

### 2.6. Variables

The variables analyzed in this paper are shown in Table 3. The Total_escape_time and Total_travel_distance were used to reflect participants’ escape performance. Eye-tracking indicators of AOI_Time_To_First_Fixation, AOI_First_Fixation_Duration, AOI_Total_Fixation_Duration, AOI_Fixation_Count, and Average_Pupil were used to measure different color safety signs’ cognitive load and notability on participants. Physiological indicators (HR, SC) were used to measure participants’ stress levels under different color safety signs.

## 3. Results and Discussion

Data collected in the experiment were preliminarily processed and exported by ErgoLAB V3.0., then IBM SPSS Statistics 22 was used for data analysis.

### 3.1. Relationship between Color_of_Safety_Sign and Escape Performance

From the one-way ANOVA results (see Table 4), there are no significances in Color_of_safety_sign, in Total_escape_time, or in Total_travel_distance (all *p* > 0.05).

Although there are no significant differences for participants’ total travel distances or total escape times in the four kinds of color of safety signs, the mean of Total_escape_time in the “Green and black” group (220.89 ± 69.04) and the mean of Total_travel_distance in the “Green and black” group (267.36 ± 61.37) are both the shortest among all the safety sign color groups (see Figure 7 and Table 4). Therefore, in terms of the escape performance, “Green and black” is the most suitable color of safety sign compared with the other three color groups. From the post-questionnaire, 75 percent of participants also rated “Green and black” as the best color to use as the safety sign, which is basically consistent with the results in this experiment.

### 3.2. Relationship between Color_of_Safety_Sign and Eye-Tracking Indicators

From the one-way ANOVA results (see Table 5), there are no significances in Color_of_safety_sign in all the five eye-tracking indicators of AOI_Time_To_First_Fixation, AOI_First_Fixation_Duration, AOI_Total_Fixation_Duration, AOI_Fixation_Count, and Average_Pupil (all *p* > 0.05).

The shorter the AOI_Time_To_First_Fixation, the easier it is to attract people’s attention and build a deep first impression [54]. From Figure 8 and Table 5, the mean of AOI_Time_To_First_Fixation in the “Green and black” group is the longest (46.74 ± 64.12), and the mean of AOI_First_Fixation_Duration in the “Green and black” group is the shortest (0.19 ± 0.16), which indicate that the “Green and black” safety sign may not attract participants’ attention in the first place, but its cognitive load is low and information processing is fast, and people could quickly understand the meaning of it.

Previous studies have shown that the larger the cognitive load, the larger the pupil diameter [55,56]. Besides, AOI_Total_Fixation_Duration and AOI_Fixation_Count increased when participants were confronted with incomprehensible contents [57]. From Figure 9 and Table 5, the mean of AOI_Total_Fixation_Duration in the “Green and black” group (1.11 ± 1.25), the mean of AOI_Fixation_Count in the “Green and black” group (6.33 ± 7.32), and the mean of Average_Pupil in the “Green and black” group (4.32 ± 2.12) are all the shortest, which indicates that there is a low cognitive load, high search efficiency, and easy information extract in “Green and black” safety sign, the same results as obtained from Figure 8.

### 3.3. Relationship between Color_of_Safety_Sign and Physiological Indicators

(1)Change of physiological indicators during virtual escape vs. baseline

According to the Tests of Normality in SPSS, from the K-S (Kolmogorov–Smirnov) results, in each four kinds of color groups, Mean_HR_baseline and Mean_HR_during_escape, all obey normal distribution (all *p* > 0.05). Mean_SC_baseline and Mean_SC_during_escape are all not normally distributed (all *p* < 0.05). Therefore, the Paired-Samples T Test and Wilcoxon Signed Ranks Test are used, respectively, to determine the changes of HR and SC, during the virtual subway fire escape compared to the baseline period.

From the Paired-Samples T Test results, the means of the HR in Color_of_safety_sign of “Green and black”, “Red and white”, and “Yellow and black” all increase very significantly (mean difference > 0, all *p* < 0.01,d (effect size) > 0.8 both in “Green and black” and “Red and white”, and d (effect size) > 0.5 in “Yellow and black”) and the mean of the HR in “Blue and white” increase significantly (mean difference > 0, *p* < 0.05, d (effect size) > 0.5), during virtual escape vs. baseline.

From the results of Nonparametric Tests, Two-Related-Samples Tests, Wilcoxon Signed Ranks Test, the median of the SC in Color_of_safety_sign of “Green and black”, “Yellow and black”, and “Blue and white” all increase very significantly (median difference > 0, *p* < 0.01), and the median of the SC in “Red and white” increases significantly (median difference > 0, *p* < 0.05), during virtual escape vs. baseline.

In a stress state, participants’ SC will increase [58]. Moreover, individual’s SNS (sympathetic nervous system) was activated under the stress state, which leads to an increased HR [59]. In this experiment, participants’ SC and HR both increased significantly in the virtual subway fire escape compared to the baseline period, which means that participants experienced a higher stress level during the fire escape. This was consistent with the previous research results [34]. Moreover, this was proved by analyzing the results of the adapted PANAS in the pre-experiment questionnaire and post-questionnaire. From the adapted PANAS in the pre-experiment questionnaire, only 20% of the participants thought they felt panic, nervous, anxious or scared, and most of these negative emotions came from the lack of knowledge about the experiment. However, from the adapted PANAS in the post-questionnaire, more than half of the participants felt panic, nervous, anxious, or scared in the virtual subway fire escape, which proves that the virtual fire escape scene brings a higher stress to participants.

(2)Relationship between Color_of_safety_sign and SC, HR

From the one-way ANOVA results (see Table 6), there is no significance in Color_of_safety_sign in the two physiological indicators of Mean_SC_increase_rate and Mean_HR_increase_rate (all *p* > 0.05).

There are no significance in the increase rates of HR and SC in the four different colors of safety signs, which may be due to the subway space being a relatively open and big space compared to the tunnel [18,46,47,60], corridor [29,61], room [48], etc. Although the size of safety signs was determined according to a 1:1 scale of the real Zijing Mountain subway station, they still looked relatively small and vague in the big subway space full of fire and smoke, which was ignored easily by participants and caused errors especially in eye-tracking data. In the post-questionnaire, participants also said that the safety signs looked too small during their entire virtual subway fire escape, which may reveal that the safety signs in the subway should be designed in a larger size in reality.

From Figure 10 and Table 6, the mean of Mean_HR_increase_rate in the “Green and black” group is the biggest (0.14 ± 0.14), and the mean of Mean_SC_increase_rate in the “Green and black” group is the second biggest (0.85 ± 0.61), very close to the biggest (0.88 ± 0.97), which indicates that participants’ stress level or physiological arousal levels are the highest in the face of “Green and black” safety signs. The reason for this may be that the safety signs in China are “Green and black” (or “Green and white”), Chinese people are more familiar with and used to this color of safety sign; thus, the other three colors reduced the immersion and reality of the virtual scene. Moreover, from the post-questionnaire, 85.4% of participants think “Green and black” means “go”or “safety”, and 76% of participants think “Red and black” conveys the meaning of “Stop”, which are consistent with other research results [62]. Meanwhile, 67% of participants think “Yellow and black” means “Warning”, and 59.4% of participants think “Blue and white” conveys the meaning of “indication”. These also show that most people take “Green and black” as the appropriate safety sign color.

### 3.4. Limitations

(1)Virtual scene

In this study, the moving speed was limited at a fixed value, and there were no NPCs in the virtual scene. The future research will provide more flexible movement speed [46], more NPCs, or other features, to make the virtual scene more real.

(2)Participants

The meaning of color is different in different cultures, even the same color has different meanings in different cultures [63]. People in different cultures also have different acceptance of colors. One study showed that Chinese and American students have different perceptions of the danger of colors [64]. In this study, participants are all from China; whether the results are applicable to other countries with different cultures remains to be further studied.

Previous research had confirmed that different age groups had different physiological abilities [65] and cognitive abilities [66] during evacuation. In this study, in order to control the influence of participants’ personal characteristics, such as gender, age, etc., on the experimental results (participants’ age, gender, vision, identity, or occupation and other personal characteristics are control variables in this experiment), participants are all undergraduate or graduate students (mean age = 22.15 ± 2.14 years) from Zhengzhou university; there is little difference in their personal characteristics, and each of the four different color groups has equal distribution in gender (male: female = 1:1) and number of participants (24 persons per color). However, the number of participants in each color group is a bit low, which may cause a non-significant difference in the four color combinations. In future studies, we will increase the subject sample groups and sizes, design more color combinations and experimental scenes, and explore the influence of different age groups and other individual characteristics on the experimental results.

(3)Single case study

In this paper, only one single case study of Zijing Mountain subway station was used, which may limit the reliability of the research results. In the future, we will carry out multiple case studies to draw more credible conclusions.

## 4. Conclusions

In this paper, VR and ET technologies are used to explore the appropriate color of safety signs from three aspects, namely, evacuation escape performance, eye-tracking indicators, and physiological indicators. For escape performance, the “Green and black” group had the best escape performance. For eye-tracking, the “Green and black” group had low cognitive load, high search efficiency, and easy information extracting and processing from safety signs. For physiological indicator changes, participants in the “Green and black” group has the highest stress level; at the same time 85.4% of participants think “Green and black” means “go“ or “safety” in the post-questionnaire, which may mean the “Green and black” color safety sign has the highest degree of immersion and reality in the virtual subway fire escape scene. In conclusion, “Green and black” is the most appropriate color for safety signs.

Although there are some limitations in this paper, this study makes some contributions. Compared with previous researches, this paper studies the effectiveness of different color safety signs from the perspective of escape performance, eye-tracking indicators, and physiological indicators, which makes the conclusions more credible. Moreover, it can improve the function of fire-fighting infrastructure and the resilience ability of subways. Meanwhile, it provides references and suggestions on risk management, emergency evacuation, etc.

## Figures and Tables

**Figure 1 ijerph-17-05903-f001:**
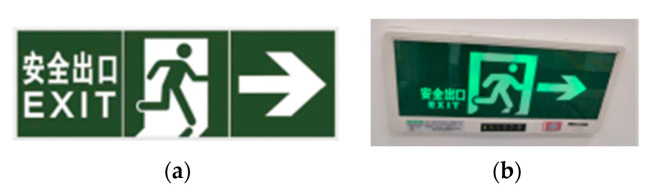
The current color combination of safety sign in China. (**a**) “Green and white”, from Chinese GB 13495.1-2015 (*Fire Safety Signs-Part 1: Signs*), and Chinese GB/T 23809-2009 (*Safety Way Guidance Systems-Setting Principles and Requirements*), etc.; (**b**) “Green and black”, the real visual effect after charging in Zhengzhou Zijing Mountain subway station and the other subway stations in China.

**Figure 2 ijerph-17-05903-f002:**
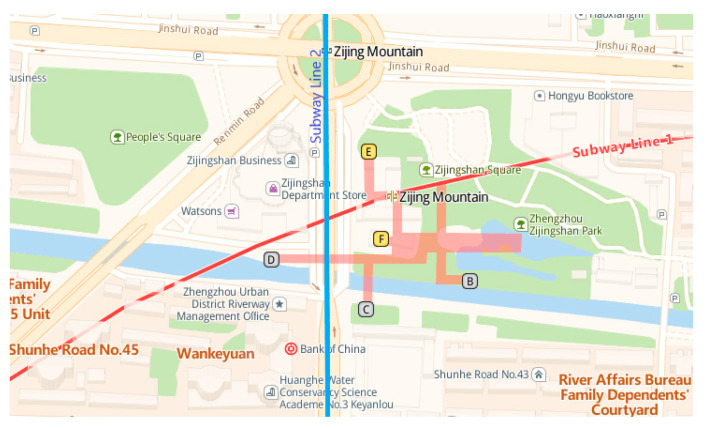
The geographical location of Zijing Mountain subway station in Zhengzhou, China (Note: This map can be accessed at the following website: https://lbs.amap.com/api/javascript-api/example/map/map-english/).

**Figure 3 ijerph-17-05903-f003:**
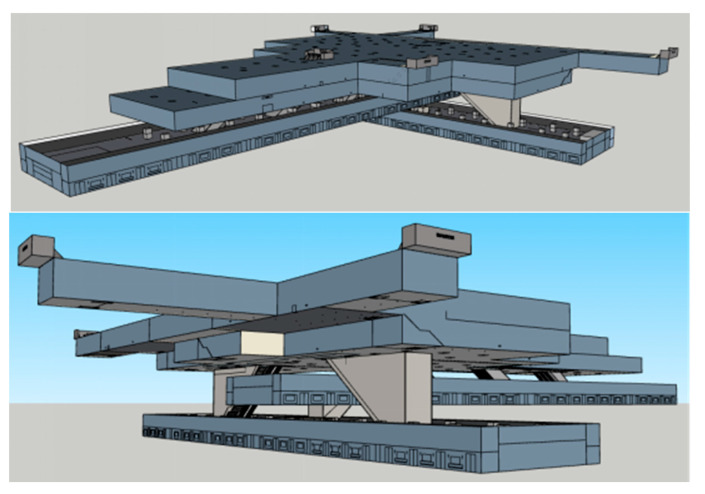
Screenshot of 3D model of Zijing Mountain subway station in SketchUp (1:1 scale of the real Zijing Mountain subway station).

**Figure 4 ijerph-17-05903-f004:**
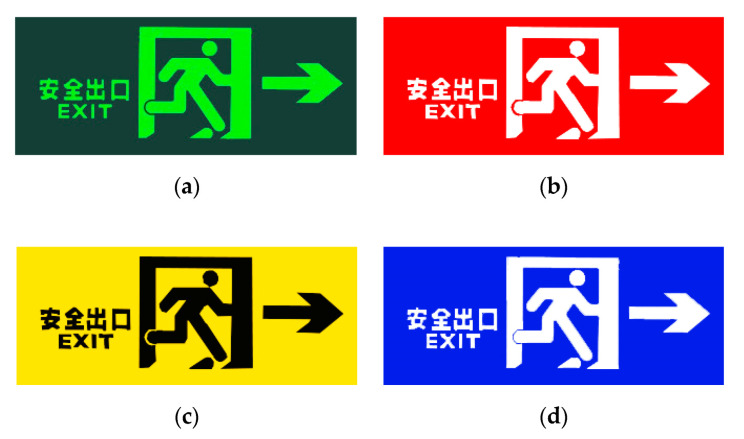
Four different colors of safety signs. (**a**) “Green and black”; (**b**) “Red and white”; (**c**) “Yellow and black”; (**d**) “Blue and white”. The size of the above safety sign pictures is 359 × 149 × 23 mm in the virtual scene, the same picture size as in the real Zijing Mountain subway station (1:1 scale).

**Figure 5 ijerph-17-05903-f005:**
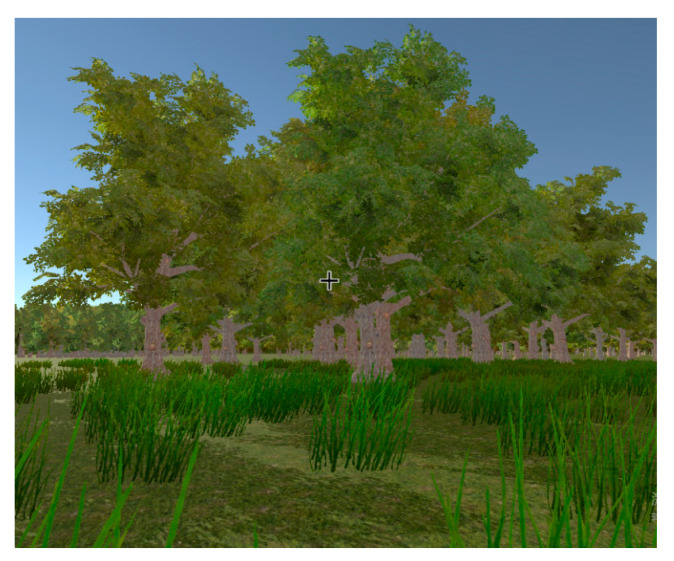
Screenshot of the virtual practice scene.

**Figure 6 ijerph-17-05903-f006:**
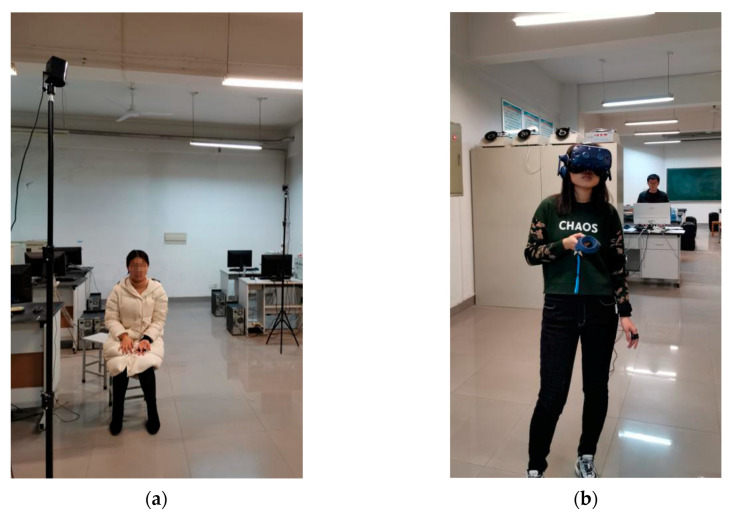
(**a**) One of the participants wore EDA sensor and PPG sensor in stable status in the baseline period; data were real-time recorded for 2 min. (**b**) One of the participants wore HTC Vive Pro Eye helmet, with a HTC trackpad in her hand; meanwhile EDA sensor and PPG sensor were put on her fingers and earlobe, during the virtual fire escape period. (**c**,**d**) Physiological data and eye-tracking data were real-time recorded by ErgoLAB V3.0, in the entire baseline period and the whole fire escape period. (**e**,**f**) When participants’ fixation points fell in the AOI (Area of Interest) of safety signs, the related eye-tracking data would be recorded; (**g**) Screenshots of the constructed virtual Zijing Mountain subway fire escape scene in Unity3D (1:1 scale of the real Zijing Mountain subway station).

**Figure 7 ijerph-17-05903-f007:**
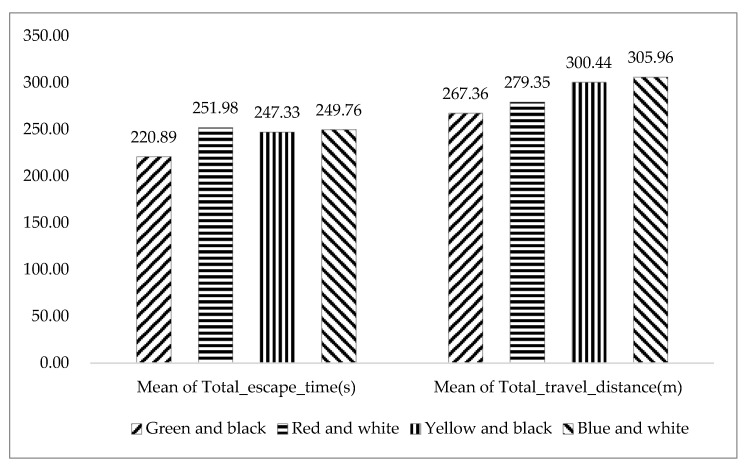
Mean of Total_escape_time and mean of Total_travel_distance in the four color groups.

**Figure 8 ijerph-17-05903-f008:**
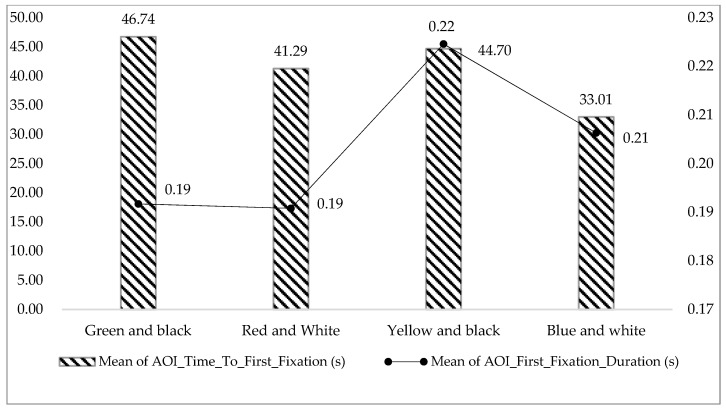
Mean of AOI_Time_To_First_Fixation and mean of AOI_First_Fixation_Duration in the four color groups.

**Figure 9 ijerph-17-05903-f009:**
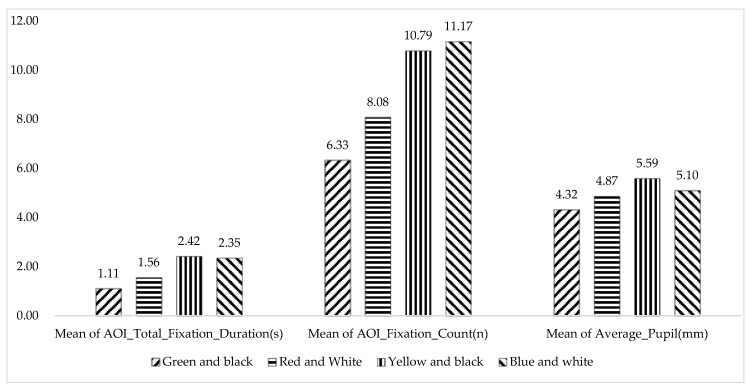
Mean of AOI_Total_Fixation_Duration, mean of AOI_Fixation_Count, and mean of Average_Pupil in the four color groups.

**Figure 10 ijerph-17-05903-f010:**
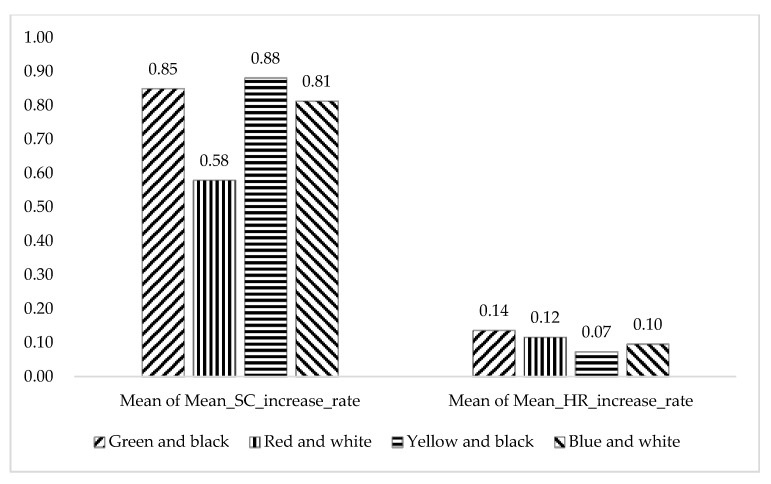
Mean of Mean_SC_increase_rate and mean of Mean_HR_increase_rate in the four color groups.

**Table 1 ijerph-17-05903-t001:** The standards referenced in this paper.

Standard Name	Standard No.	Notes
*Fire Safety Signs-Part 1: Signs* [38]	GB 13495.1-2015	China National Standard
*The Passenger Service* *Signs for Urban Rail Transit* [39]	GB/T 18574-2008	China National Standard
*Safety Way Guidance Systems-Setting Principles and Requirements* [40]	GB/T 23809-2009	China National Standard
*Safety Signs and Guideline for the Use* [41]	GB 2894-2008	China National Standard
*Safety Colors* [42]	GB 2893-2008	China National Standard
*Standard for Fire Safety and Emergency Symbols* [43]	NFPA (National Fire Protection Association) 170-2018	American National Standard

**Table 2 ijerph-17-05903-t002:** The hardware and software used in the experiment.

	Name	Description
Hardware	HTC Vive Pro Eye 2.0 (Hongda Communications Co. LTD, Shanghai, China)	The virtual reality Head-Mounted Display (HMD), containing two 3.5-inch 3K AMOLED displays, each with a resolution of 1440 × 1600, a maximum field of view Angle of 110°, and a screen refresh rate of 90 HZ
	PPG (Photoplethysmography) sensor (Kingfar International Inc., Beijing, China), EDA sensor (Kingfar International Inc., Beijing, China)	The real-time physiological data acquisition device, recording participants’ HR and SC (Skin Conductance) data
	HTC trackpad (Hongda Communications Co. LTD, Shanghai, China)	Participants moving or making turns with it in the virtual subway fire escape
	Unity3D VR Plugin data synchronization interface adapter (Kingfar International Inc., Beijing, China)	Data in VR scene were transmitted and recorded synchronously by this device
	CPU (i5-8400), GPU (GTX 1060)	The computer operating environment in the virtual experiment
Software	SketchUp (Version of 2018pro, Trimble, Sunnyvale, CA, USA)	The virtual reality scene model was created by it according to the real subway station of Zijing Mountain in Zhengzhou, China, with 1:1 scale
	Photoshop CS6 (Adobe Systems Incorporated, San Jose, CA, USA)	Some images were designed by it as textures in Unity3D
	Unity3D (Version of 2019.2.11f1, Unity Technologies, San Francisco, CA, USA)	The created model was imported into it for the interactive setting of scene functions
	ErgoLAB V3.0 man-machine-environment synchronous cloud platform (Kingfar International Inc., Beijing, China)	This software was used for real-time physiological data acquisition and processing, including ErgoLAB wearable wireless physiological recording module (PPG, EDA) and virtual reality eye movement tracking module (Tobii VR)
	IBM SPSS Statistic 22 (IBM Corporation, Armonk, NY, USA)	This software was used for data analysis

**Table 3 ijerph-17-05903-t003:** The variables analyzed in this paper.

Variable Name	Meaning	Unit
Total_escape_time	Total time that participants took to escape to the ground exits from the escape starting point in the −4F	s
Total_travel_distance	Total moving distance that participants travel to escape to the ground exits from the escape starting point in the −4F	m
AOI_Time_To_First_Fixation	Time to first fixation at the safety sign in AOI (Area of Interest)	s
AOI_First_Fixation_Duration	The fixation duration time of the first fixation point at the safety sign in AOI	s
AOI_Total_Fixation_Duration	Total fixation duration time for all the fixation points at the safety signs in AOI	s
AOI_Fixation_Count	Total fixation number (count) for all the fixation points at the safety signs in AOI	n (number)
Average_Pupil	The average pupil diameter of left and right eyes during the fixation at all the fixation points at the safety signs in AOI	mm
increase_rate	The growth rate of the variable during escape vs. baseline	dimensionless
Mean_SC	The mean of the Skin Conductance during escape or baseline	μs
Mean_HR	The mean of the Heart Rate during escape or baseline	bpm (beats per minute)
Color_of_safety_sign	Four different color combinations of safety signs: “Green and black”, “Red and white”, “Yellow and black”, and “Blue and white”	-

**Table 4 ijerph-17-05903-t004:** One-Way ANOVA of participants’ escape performance (Grouping Variable: Color_of_safety_sign).

Color_of_Safety_Sign	Total_Escape_Time (Mean ± SD)	Total_Travel_Distance (Mean ± SD)
Green and black (*n* = 24)	220.8875 ± 69.03976	267.3625 ± 61.37007
Red and white (*n* = 24)	251.9750 ± 85.60563	279.3542 ± 94.52431
Yellow and black (*n* = 24)	247.3250 ± 112.72859	300.4417 ± 127.57933
Blue and white (*n* = 24)	249.7583 ± 114.68886	305.9625 ± 130.72830
Sig.	0.66	0.567

**Table 5 ijerph-17-05903-t005:** One-Way ANOVA of participants’ eye-tracking indicators (Grouping Variable: Color_of_safety_sign).

Color_of_safety_sign	AOI_Time_To_First_Fixation (Mean ± SD)	AOI_First_Fixation_Duration (Mean ± SD)	AOI_Total_Fixation_Duration (Mean ± SD)	AOI_Fixation_Count (Mean ± SD)	Average_Pupil (Mean ± SD)
Green and black (*n* = 24)	46.7446 ± 64.12274	0.1917 ± 0.16433	1.1063 ± 1.25075	6.3333 ± 7.32279	4.3167 ± 2.12531
Red and white (*n* = 24)	41.2896 ± 68.69189	0.1908 ± 0.16686	1.5621 ± 1.53618	8.0833 ± 7.76232	4.8696 ± 1.63910
Yellow and black (*n* = 24)	44.6971 ± 61.58409	0.2246 ± 0.15959	2.4242 ± 3.69194	10.7917 ± 14.50031	5.5913 ± 1.37096
Blue and white (*n* = 24)	33.0075 ± 42.15658	0.2063 ± 0.14467	2.3504 ± 3.21087	11.1667 ± 13.51864	5.1004 ± 1.79847
Sig.	0.865	0.87	0.249	0.397	0.095

**Table 6 ijerph-17-05903-t006:** One-way ANOVA of participants’ physiological indicators (Grouping Variable: Color_of_safety_sign).

Color_of_Safety_Sign	Mean_SC_Increase_Rate (Mean ± SD)	Mean_HR_Increase_Rate (Mean ± SD)
Green and black (*n* = 24)	0.8496 ± 0.60927	0.1367 ± 0.13586
Red and white (*n* = 24)	0.5796 ± 0.88858	0.1167 ± 0.11461
Yellow and black (*n* = 24)	0.8813 ± 0.96647	0.0738 ± 0.11912
Blue and white (*n* = 24)	0.8133 ± 1.10886	0.0963 ± 0.17126
Sig.	0.654	0.428

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
