# Peer review of "The Physiological Experimental Study on the Effect of Different Color of Safety Signs on a Virtual Subway Fire Escape—An Exploratory Case Study of Zijing Mountain Subway Station"

_ijerph, 2020, doi:10.3390/ijerph17165903_

Round 1

Reviewer 1 Report

The authors have done a good job in this revision. All my previous major concerns have been addressed. However, there are still numerous minor typos and mistakes. In addition, the revision on statistical analysis has come to my attention. Thus, here I raise one more major concern.

Major concern:

  1. Line 248-256 - The authors used Wilcoxon signed-rank test here to test "mean" difference. It sounds suspicious: (1) the author did not show numerical results of both tests; (2) Wilcoxon signed-rank test is the nonparametric test equivalent to the Dependent t-test, a.k.a. Paired-sample t-test in SPSS. Instead of using the same test, why did the authors change from Paired-sample t-test to nonparametric Wilcoxon test? Please show evidence that the normality of the samples cannot be met so that a nonparametric test is really a necessity here. Otherwise, it is inappropriate to use a nonparametric test here when your sample's distribution can be approximated with normal distributions, vice versa, if the distribution of samples in Paired-sample t-test does not met the assumption on normality either, the authors should use nonparametric test for both tests here. The point is that the authors should check the assumptions of these tests first before using them and show the evidence to justify; (3) A Wilcoxon signed-rank test is a nonparametric test that can be used to determine whether two dependent samples were selected from populations having the same distribution. As a nonparametric test, talking about "means" would be problematic because "mean" is a parameter of normal distributions (therefore, parametric). Given that the sample distributions does really not follow normal distributions (i,e. using Wilcoxon test is correct), "medians" (instead of "means") should be used to report the test result; (4) Finally, as we all know, dependent t-test is sensitive to effect sizes. Effect sizes are important because whilst the dependent t-test tells you whether differences between group means are "real", it does not tell you the size of the difference. The authors should also calculate the effect sizes, instead of just showing p values. This is more convincing.

To name a few typos and mistakes in the MS:

  1. Line 14 - "Ninety-six". In scientific writing, if a counting number is larger than 12, you don't write it in words.
  2. Line 16 - there should be a space between "data" and the brackets.
  3. Line 18 - Check the format of the word "Color_of_safety_sign". Why label the variables like this? Same thing happens again from Section 2.5. You don't have to follow the same label as shown in SPSS.
  4. Line 31 - Subway is not one of the vehicles (it is a "type of vehicles" or "transportation modes")! and also, it is one of the main types of urban transportation modes not because of its large volume! I think the size does not matter, does it? it is more because of its capability to accommodate "large volume of passengers", isn't it? The authors should improve the accuracy of their words.
  5. Line 43 - The authors missed the surname of the author of the reference [7] in the reference list. Please check other references as well to avoid such mistakes.
  6. Line 44 - space missing between "system[8],improve....". Same problem at Line 47, 48, 52, and many other spots throughout the MS.
  7. Line 146 - Please give credit and citation to the base map you used.

I would like to see the authors being more rigorous about details this time.

Author Response

we are very grateful for all your opinions and suggestions, thank you very much for your hard work and valuable help!

Reviewer 2 Report

This paper provides the results of an experimental study on the effect of different color of safety signs on several escape performance variables (ie, total scape time, total travel distance) and physiological and eye-tracking data (ie, skin conductance, heart rate, time to first fixation, first fixation duration, total fixation duration, etc).

I think this article has the same limitations as the original version.

* The experiment is based on virtual reality using only 96 students, with only one case study (a subway station in China) and four possible color combinations.

* Each participant only performs the experiment once (using only one of the safety sign color combinations); the personal characteristics of the participants could influence the small differences identified among colors; the number of participants is very low (24 per color; the age range is also reduced, etc.

* The differences on colors combinations are not statistically significant.

From my point of view there are only two minor improvements: the English language along the document; and the acknowledge of some limitations. However, the authors do not acknowledge the limitation in the number of participants, that it is the most important one.

Author Response

We are very grateful for all your opinions and suggestions, thank you very much for your hard work and valuable help!

Reviewer 3 Report

The reviewer thanks the authors for the changes made in the document.

Author Response

Thank you very much for your valuable suggestions and comments. We really appreciate your last help. Wish you happy every day!

Round 2

Reviewer 1 Report

Thank you for all your efforts addressing additional comments. The authors have corrected the mistakes and the revision is now satisfactory.

Author Response

Thank you very much for your valuable suggestions and comments before! We really appreciate your help! Wish you happy every day!

Reviewer 2 Report

Thank you for including the main limitation in the discussion section.

Author Response

Thank you very much for your valuable suggestions and comments before! We really appreciate your help! Wish you happy every day!

This manuscript is a resubmission of an earlier submission. The following is a list of the peer review reports and author responses from that submission.

Round 1

Reviewer 1 Report

The paper describes the research conducted to find out the best colour for evacuation escape signals in an emergency. The authors used virtual reality, eye tracking and psychological data in real-time recording to learn the reactions of the participants to different colours signals.

The paper fits the scope of the journal and shows interesting research. However, in the opinion of this reviewer, it should be not published in its present form, so that, some comments and concerns are included.

In Section 2.1, lines 78 to 92, the information included is confusing. It could be clearer if it were displayed in another format, as in a table.

In Section 2.2, the authors described the participants in the experiment. They were all between 18 to 28 years old. In the opinion of this reviewer, other groups of people could have been included. Furthermore, they had a normal or corrected-to-normal vision, as well as normal colour vision. It would be interesting to compare the results between people with normal vision y people with vision problems or abnormal colour vision, following the principles of “Design for All”.

Section 3 is a little confusing. Some tables or bullet points may be used to identify the variables.

The conclusions in Section 4 are also confusing, especially in the first paragraph which is very long and not clear. The Section could include some future research, such as the comparison with people of other ages or characteristics.

Reviewer 2 Report

This paper provides the results of an experimental study on the effect of different color of safety signs on several escape performance variables (ie, total scape time, total travel distance) and physiological and eye-tracking data (ie, skin conductance, heart rate, time to first fixation, first fixation duration, total fixation duration, etc).

The experiment is based on virtual reality using 96 students, with only one case study (a subway station in China) and for possible color combinations. Each participant only performs the experiment once (using only one of the safety sign color combinations).

This study presents methodological and results limitations.

From the point of view of results: The differences for the colors combinations are not statistically significant.

From the point of view of the methodology: The study is based on only one case study, each participant only evaluates one of the colors (the personal characteristics of the participants could influence the small differences identified among colors), the number of participants is very low (24 per color), the age range is also reduced, etc.

The paper also presents limitations in the language and in almost all the sections of the manuscript (literature review, methodology, data description, results and conclusions, etc.)

Reviewer 3 Report

This paper studies different safety sign designs from a series of physiological experiments, using cutting-edge ergonomic and digital tools such as VR. The reviewer found this study really of interest and has potential values for multiple communities, such as engineering management, risk management, and automation in construction. However, the MS needs to be revised before considering for publication. Please find the following specific comments for the revision. The reviewer hopes the authors find them useful.

  1. There are many minor grammar mistakes. For example, the missing and misuse of the article “a”, “an”, and “the”. Also, there are numerous typos of formatting of lines and paragraphs. For instance, please be careful with the spacing between punctuations and lines. I would suggest authors to proofread the whole MS carefully in revision.
  2. The abstract contains too much details about the specific tools used. Instead of doing so, I suggest to include more details of experimental design, results, and implications (rather than saying what specific tools were used). The authors should clearly indicate that what implications and managerial lessons could be drawn from this study, and also, to whom these implications and lessons might have positive and valuable impacts.
  3. The introduction section is overall OK, but two very important pieces of information are missing. (1) a substantial body of literature review. What has been done in previous studies? What is the state-of-the-art knowledge on this topic? Who has done what in this research field? What are the research gaps? All those these mentioned aspects should be carried out in a more detailed way (Please read more relevant previous work and cite them. This is also important for the second piece in the following comment) This literature review can be either done in the introduction section or formed as a separate section after the introduction section.
  4. The second piece is: (2) the major contributions of this paper. The authors could end this section with a quick and clear summary of the contributions to the wider community. Why is this study valuable to others? What are the take-home messages you want your readers to learn? A few bullet points should be enough for summarizing the main contributions. This is fairly important for readers.
  5. In the literature review, it would be good to add more literature from topics of “fire risk prevention in subway systems” and a wider aspect of “resilience in infrastructure systems” for the sake of a wide range of readers. Because the reviewer can see that this study has potential values for multiple communities, such as engineering management, risk management, and automation in construction, etc. But the authors did not highlight such potential values in the beginning of the paper. Thus, please include a wider literature review and highlight the contributions. To name a few really excellent previous works:

    For fire risk prevention:

  • Zhang, L., Wu, X., Liu, M., Liu, W. and Ashuri, B., 2019. Discovering worst fire scenarios in subway stations: a simulation approach.Automation in Construction99, pp.183-196.
  • Zhang, L., Liu, M., Wu, X. and Zhong, J.B., 2014. Risk analysis of crowded stamped accident in subway station based on DEA method.J Civ Eng Manage31(4), pp.76-82.
  • Zou, P.X. and Li, J., 2010. Risk identification and assessment in subway projects: case study of Nanjing Subway Line 2.Construction Management and Economics28(12), pp.1219-1238.
  • Mu, S., Cheng, H., Chohr, M. and Peng, W., 2014. Assessing risk management capability of contractors in subway projects in mainland China.International Journal of Project Management32(3), pp.452-460.

    For resilience in infrastructure systems:

  • Tang, J., 2019. Assessment of Resilience in Complex Urban Systems.Industry, Innovation and Infrastructure.
  • Gernay, T., Selamet, S., Tondini, N. and Khorasani, N.E., 2016. Urban infrastructure resilience to fire disaster: An overview.Procedia engineering161, pp.1801-1805.
  • Bobylev, N., 2007. Sustainability and vulnerability analysis of critical underground infrastructure. InManaging Critical Infrastructure Risks (pp. 445-469). Springer, Dordrecht.
  1. Section 2.1 – need substantial amount of citations for those standards.
  2. Page 4 Line 116 – could add a geographical figure to point out where the Zijinshan station is on the Zhengzhou metro system.
  3. The reviewer would suggest to convert the whole section 2.3 into a summary table rather than fragmented paragraphs as shown in the current form of the MS.
  4. The resolution of the figure 5 is too low, please consider optimizing it.
  5. Page 7 Line 187-202 – the content here needs to be re-organized in a logical manner. Besides, the information here does not belong to “Result”.
  6. Section 3.1-3.3 – the way of representing the results should be optimized, now the information is presented in a really messy way and non-logic way. In addition, authors should provide more in-depth discussions about the results. For example, what are the results indicating in the practice? are there any implications that can related to wider real-world phenomena? Things like those aspects.
  7. The reviewer would suggest authors adding some discussion on the limitations of this study. For example, bias control? error estimation? constraint on participants’ age (all youngsters, how about elderly people? Is there any implication for them from this study?), and can the findings in this study be generalized and transferable to other cities or countries?
  8. Finally, please revise the whole story line. The story line needs to be presented in a clearer way, especially the section 2 and 3.